# Peer Support Provided by People with Intellectual and Developmental Disabilities: A Rapid Scoping Review to Develop a Toolkit for Inclusive Research

**Beth Pfeiffer** [1,*] **, Taye Hallock** [2]**, Luke Tomczuk** [1] **and Jessica Kramer** [3]

1 REACH Lab, Department of Health and Rehabilitation Sciences, Temple University, 1913 North Broad Street, Philadelphia, PA 19122, USA; luke.christopher.tomczuk@temple.edu
2 Institute on Disability, Temple University, 1755 N 13th Street, Philadelphia, PA 19122, USA; taye.hallock@temple.edu
3 Department of Occupational Therapy, University of Florida, 1225 Center Drive, Gainesville, FL 32611, USA; jessica.kramer@phhp.ufl.edu
* Correspondence: bpfeiffe@temple.edu

**Abstract:** Inclusive research teams actively engage people with intellectual and developmental disabilities at all stages of research development, implementation, and dissemination. There is a dearth of research that specifically addresses the use of peer support in research engagement, yet research using peer support may provide a useful framework for engagement in inclusive research teams. A rapid scoping review was completed following the reporting guidelines for PRISMA-SCR. The scoping review identified five peer support roles (communication, sharing experiences, helping peers to learn, peer development, and creating a welcoming environment) and two types of support and accommodation for peer supporters (individual and environmental). The findings of the rapid scoping review aided in developing key sections of a Peer Support Toolkit to help people with intellectual and developmental disabilities engage in research to create more inclusive research teams and research that is informed directly by the needs of people with lived experience. The scoping review and toolkit were completed by an inclusive team.

**Keywords:** intellectual and developmental disabilities; peer support; inclusive research; scoping review; rapid scoping review





## 1. Introduction

Partnering with people with intellectual and developmental disabilities in research (inclusive research) can ensure that research is important and meaningful. Inclusive research often leads to increased interest in participation and greater acceptance of results, as well as a process that is empowering to inclusive research team members (Domecq et al. 2014; Dudley et al. 2015; Harrison et al. 2019; Sheridan et al. 2017; Walmsley et al. 2018). In the last twenty years, leaders in inclusive research have established methodological best practices for decision-making processes, accessible data collection and analysis, and dissemination (Frankena et al. 2019). Yet, there is still a pressing need for quality approaches to training and building the capacity of research collaborators with intellectual and developmental disabilities that are aligned with the key tenets of inclusive research (Embregts et al. 2018; Garratt et al. 2022; Milner and Frawley 2019). Peer support can be a useful approach for facilitating engagement for people with intellectual and developmental disabilities in the research process in a way that is performed 'with' and 'by' people with disabilities (Kramer et al. 2013, 2023, 2018; Milner and Frawley 2019; Strnadová et al. 2014; Tavecchio et al. 2019).

The (Developmental Disabilities Assistance and Bill of Rights Act 2000) of the United States of America defines a developmental disability as a condition that is attributable to a mental and/or physical impairment, manifested before the age of 22 years, long-term,

results in significant limitations in multiple areas of functioning, and requires specialized supports (Developmental Disabilities Assistance and Bill of Rights Act 2000). A broad range of specific diagnostic conditions are encompassed by the category of developmental disabilities, including Autism Spectrum Disorder, intellectual disability, and cerebral palsy (Boyle et al. 2011). The term developmental disability is often used interchangeably with other labels, like neurodevelopmental conditions. However, Intellectual Developmental Disorder is a more specific condition that falls under the umbrella of developmental disabilities. People with developmental disabilities and/or their families in the US use a variety of terms to describe their identity, such as self-advocate, autistic or neurodiverse, person with lived experience, and person with special needs. However, there is no clear consensus, and terminology may be regional- or diagnostic-specific, and it may vary depending on the context in which it is used (Autistica 2023; Self Advocates Becoming Empowered 2023; Self-Advocacy and Leadership 2023). Given that the population described above is a highly heterogeneous group, we use the term intellectual and developmental disabilities in this manuscript to reflect all of these individuals.

For people with intellectual and developmental disabilities, there is growing evidence that peer support improves outcomes in areas such as independent living, socialization and relationships, and employment (Brady et al. 2016; Causton-Theoharis 2010; Chan et al. 2009; Griffin et al. 2016). Peer support is an organized method of providing formal or informal support and is founded on the concept that a peer with lived experience is poised to better understand the unique perspective of another person with a similar experience (Bazzano et al. 2009; Frawley and Bigby 2014; Power et al. 2016; Schwartz et al. 2020; Pfeiffer et al. 2021). Similarly, engagement in research leads to transformative change for people with intellectual and developmental disabilities in their professional, personal, and community lives (Herer and Schwartz 2022; Hopkins et al. 2022; Zaagsma et al. 2022). Current literature on peer support focuses mainly on the effectiveness of a specific peer support intervention on a targeted outcome (Weidle et al. 2006); yet, there is a dearth of research that has systematically examined the use of peer support in research engagement. Still, existing peer support research, conducted outside of a specific inclusive research approach, may provide a useful framework for integrating peer support as a mechanism for engagement in inclusive research teams.

People with intellectual and developmental disabilities have multiple identities and characteristics that they can leverage as peer mentors. This may include their gender identity and expression, cultural and linguistic backgrounds, and nationality (Wehmeyer et al. 2017). In this paper, we specifically focus on peer support provided by people with intellectual and developmental disabilities. This is a purposeful challenge to the over-representation of non-disabled, same age "peers" in peer support literature that includes people with intellectual and developmental disabilities (Płatos and Wojaczek 2018; Travers and Carter 2022). Peer support builds on the theory that shared lived experience and reciprocal relationships provide a mechanism for change (Substance Abuse and Mental Health Services Administration (SAMHSA) (2015)). The use of peers who do not share the lived experience of disability in educational, vocational, and social interventions eliminates a significant component of the proposed mechanism of change and disregards the lived experience, expertise, and capacity of persons with intellectual and developmental disabilities (Bigby et al. 2014; Walmsley et al. 2018). Again, as aligned with the key tenets of inclusive research (Embregts et al. 2018; Garratt et al. 2022; Milner and Frawley 2019), conceptualizing peer support by experienced researchers with intellectual and developmental disabilities is an opportunity to further advance inclusive research methodology.

We conducted a rapid scoping review and developed a Peer Support Toolkit for collaborative research teams with the primary aim of identify strategies and expanding the use of peer support as an inclusive research method. The purpose of this rapid scoping review was to identify the key components of peer support including roles, strategies, and supports, provided by individuals with intellectual and developmental disabilities that could translate into methods for research engagement. The review included a range of

different study designs and methods across the literature that used peer support strategies and interventions provided by and for people with intellectual and developmental disabilities. Results were integrated into a Peer Support Toolkit, with the collaboration of a team of researcher with intellectual and developmental disabilities, to support the engagement of people with lived experience in research (Pfeiffer et al. 2021).

## 2. Methods

The review followed the reporting guidelines for PRISMA-SCR (Tricco et al. 2018). A rapid scoping review is a recommended method when including a range of study designs and methods across the published and gray literature (Sucharew and Macaluso 2019). This method was suited to our study aim given the growing, but comparably limited, literature on the topic of peer support provided by individuals with intellectual and developmental disabilities. Further, a rapid review aligned with the overall study goal to quickly translate the existing knowledge into a toolkit. This allowed us to extract strategies and components across a variety of research literature specific to peer support by and for individuals with intellectual and developmental disabilities. The research team consisted of two project leads with extensive experience with inclusive research teams, a stakeholder, research staff with intellectual and developmental disabilities, and two additional research staff.

### 2.1. Search

To identify studies to include or consider for this rapid scoping review, the review team collaborated with a medical librarian to develop detailed and systematic search strategies for each database (Bramer et al. 2018). Details of the full search strategy are provided at http://hdl.handle.net/20.500.12613/4663 (accessed on 21 December 2023) (Roth et al. 2020), which is a freely available repository for sharing and archiving a range of scholarly works, including search processes for reviews. Specific search methods and terms for each database are available in metadata available at this site for replication purposes. Developed for PubMed (NLM) initially, the search was then translated to ERIC (EbscoHost), CINAHL (EbscoHost) and PsycInfo (EbscoHost) using a combination of keywords and subject headings. A gray literature search included dissertations and theses within Dissertations & Theses Global (Proquest). Included articles were written in English and published between 2005 and 2020. Given the relatively recent emergence of peer support in the field of intellectual and developmental disabilities, these dates were appropriate and helped to ensure the rapid pace of our review. The search excluded studies with children and only included articles with participants aged thirteen years and over. The age range was set at thirteen years and above to ensure the inclusion of articles focused on transition age youth and the young adult population with intellectual and developmental disabilities, as there is a subset of the literature that specifically focuses on peer support within this age range (Ryan et al. 2016; Weidle et al. 2006).This allowed the team to find literature that included information that could be translated into peer support strategies for adults with intellectual and developmental disabilities on inclusive research teams. Researchers completed the final search on 24 November 2020. The search resulted in 3154 studies. Endnote X.7 identified 267 duplicate studies and omitted these for the deduplication of records, and 2887 references were eligible to screen for inclusion (see Figure 1).

### 2.2. Screening and Review

The researchers used Covidence, a web-based collaboration software platform (Covidence Systematic Review Software 2020), to manage the screening and review process. The research team determined the inclusion and exclusion criteria prior to the review. The inclusion criteria were as follows: (1) a study population of people with intellectual and developmental disabilities aged 13 years or older (including intellectual disability, autism, cerebral palsy, and other developmental disabilities); (2) the description and use of intentional peer support interactions; (3) peer support provided by a person with an intellectual and developmental disability; and (4) articles published in English. The researchers

excluded articles if (1) people without an intellectual and developmental disability provided peer support (e.g., no diagnosis, spinal cord injury, specific learning disabilities, ADHD) and/or (2) publications were conference presentations/abstracts, book chapters, or websites.

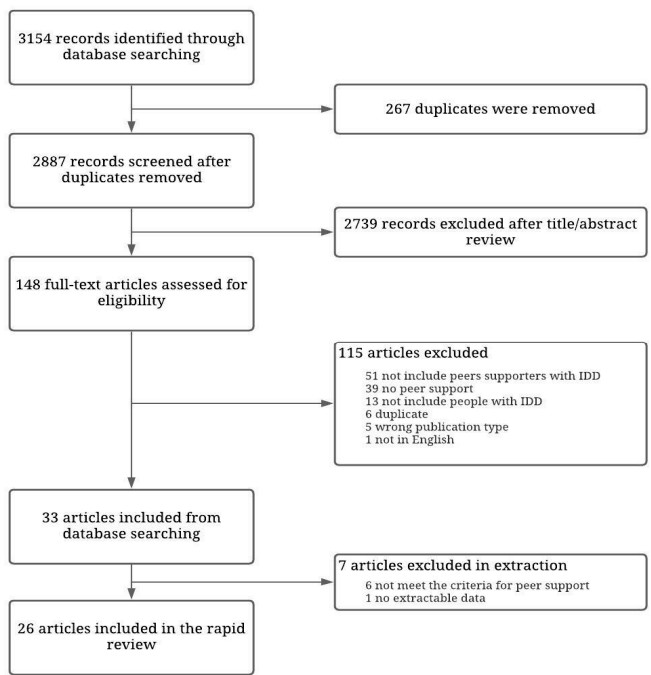

**Figure 1.** *PRISM* flow diagram. Note that this figure provides a *PRISM* flow diagram of the article's search and review process in each phase of the scoping review.

A total of five research team members were involved in the review process. One of the team members and authors was a person with an intellectual and developmental disability who identifies as an autistic adult. He was involved in the screening of abstracts and the full-text review of articles, as well as resolving conflicts between other reviewers. Two blinded and independent reviewers screened each study by title and abstract. If there was a conflict between reviewers, a third reviewer was involved in a consensus discussion with the two initial reviewers. Title and abstract screening resulted in the exclusion of 2739 records. The research team repeated this process for full-text article screening and article selection. The research team reviewed 148 full text articles and excluded 115 for the following reasons:

1.  Did not include people with intellectual and developmental disabilities (n = 13). Articles were excluded that did not include people with intellectual and developmental disabilities over the age of 13 years. Reviewers excluded articles that focused on other non-intellectual and developmental disabilities (i.e., specific learning disabilities such as ADHD, dyslexia).
2.  Did not include any identifiable peer support component (n = 39). Articles were excluded that did not identify peer support or any of its components in the full text, including those articles that referred only to naturally occurring peer relationships or friendships. Researchers also excluded articles that did not describe an intentional interaction between peers, such as those using pre-recorded videos of a peer modeling a target behavior or outcome.
3.  Did not include peers with intellectual and developmental disabilities (n = 51). Articles were excluded that did not have a peer with an intellectual and developmental disability (e.g., a typically developing peer) delivering the peer support strategy or intervention.

4.     Wrong publication type (n = 5). The research team excluded conference abstracts and presentations, book chapters, and websites.
5.     Not in English (n = 1). Articles were excluded that were not written in English, as researchers did not have access to confirm accurate translation.
6.     Duplicates (n = 6). Duplicate articles identified by reviewers were excluded.

Following a full-text review, 33 articles that described peer support for people with intellectual and developmental disabilities were retained. During the extraction process, researchers eliminated another seven articles, as one article did not have any extractable data (n = 1) and the remaining six did not meet the criteria for peer support (n = 6), as there was no *intentional* interaction between peers (e.g., those using pre-recorded videos of a peer modeling a target behavior or outcome). The final review included 26 full-text articles.

A thematic analysis process was used when extracting the data. Two research team members completed line-by-line coding of text initially from three randomly chosen articles to identify themes and categories to guide the extraction of information on peer support. Any new categories that emerged during data extraction were added after discussion and consensus from research team members. One person on the research team extracted all descriptions of peer support from each article. A second reviewer reviewed and validated the extracted information, revising or adding to the extracted data as needed. The full research team discussed questions about discrepancies in reaching consensus about the final content. Researchers categorized data extracted on peer support from each article as (1) terms used to describe peer supporters, (2) definitions of peer support, (3) descriptions of what peer supporters did, and (4) descriptions of assistance or training provided for peer supporters. The project leads then collaboratively coded these descriptions into key components of peer support, peer supporter roles and strategies, as well as supports and accommodations. These coded data were shared and discussed by an advisory board which included people who had lived experience with intellectual and developmental disabilities (n = 4), family and caregivers of people with intellectual and developmental disabilities (n = 3), and researchers with an intellectual and developmental disability (n = 3) to identify important themes and content for using peer support in inclusive research teams.

The aim of this review was to identify ways that peer support is provided by people with intellectual and developmental disabilities for the purposes of translating those methods into research engagement, not to evaluate the effectiveness of those strategies. The research team wanted to include a full range of study designs that incorporated peer support. Therefore, a critical appraisal of each article was not included.

## 3. Results

Three main themes were identified during the data extraction and analysis process including (1) key components of peer support; (2) peer supporter roles and strategies; and (3) supports and accommodations for peer supporters. The results provide descriptions of these themes, along with an overview of the article characteristics. Characteristics of the articles including the type of article, participants, design, and terminology used for peer support are identified to provide context for the interpretation and application of data.

### 3.1. Article Characteristics

We identified 26 articles that described peer support provided by people with intellectual and developmental disabilities. See the Table S1 for the article's details. Out of the 26 articles, eight focused on peer support provided by and for individuals with Autism Spectrum Disorders (ASD), ten on people with an intellectual and developmental disability other than ASD, and eight on a combination of a variety of developmental disabilities (including ASD and intellectual and developmental disabilities). The age range of peers with intellectual and developmental disabilities and those they supported across studies was from 13 to 71 years. Terms used to describe the peer supporter varied considerably across articles and included peer mentor (n = 8), peer tutor or educator (n = 5), peer facilitator (n = 3), and peer coach (n = 2). Ten articles used other terms to characterize peer

support, such as coach, advocate, and ambassador (see Table S1). The various terms used to describe support provided by peers can guide research engagement and provide options for inclusive research teams for language that best aligns with their teams' preferences and philosophy.

The types of studies and articles varied and included twelve qualitative studies, eight single group intervention studies, three program evaluations, one case study, one editorial with included supporting data, and one two-group pre-test–post-test study using a randomized control design with a wait-list control group. A range of qualitative methods were used to elicit information about the peer support process including focus groups, interviews, observations, and self-reflections. The thematic coding was performed either solely by the research team or sometimes cooperatively with the participants.

*3.2. Key Components of Peer Support*

A primary theme that emerged from the results of the data extraction and analysis process was key components of peer support (see Table 1). These key components help to identify the common core characteristics of peer support, as conceptualized and reported in the literature. Establishing these key components reveals the hypothesized mediators that lead to the observed outcomes and benefits of peer support. In the context of inclusive research teams, peer support may need to incorporate these key components to ensure similar benefits.

The most commonly identified component of peer support was having a shared lived experience (n = 15) (Bazzano et al. 2009; Bertilsdotter Rosqvist 2019; Carley 2018; Crane et al. 2021; Dudley et al. 2015; Eisenman et al. 2014; Frawley and Bigby 2014; Kramer et al. 2018; Schwartz et al. 2020; Singh et al. 2011; Strnadová et al. 2014; Williams and Porter 2017). This was often described in the context of their diagnosis and the similar experiences that the peer and peer supporter shared. The second-most identified component was self-efficacy and self-advocacy (n = 11) (Bazzano et al. 2009; Borisov 2009; Eisenman et al. 2014; Frawley and Bigby 2014; Marks et al. 2019; Nind et al. 2021; Power et al. 2016; Strnadová et al. 2014; Weidle et al. 2006; Williams and Porter 2017; Wright et al. 2019). Self-efficacy refers to a person's belief in their own capacity, whereas self-advocacy is the ability to advocate for oneself or their views. These were described as constructs which a peer supporter facilitated in the peer support relationship. Reciprocity was identified as core characteristics of peer support in a number of the articles (n = 9) (Borisov 2009; Frawley and Bigby 2014; Kramer et al. 2018; Nind et al. 2021; Ryan et al. 2016; Strnadová et al. 2014; Williams 2015; Williams and Porter 2017; Wright et al. 2019). The construct of reciprocity was described as the mutual benefit of both the peer and peer supporter within the peer support relationship.

*3.3. Peer Support Roles and Strategies*

A related but different theme that was identified in the data extraction and analysis process was *peer support roles and strategies.* This theme describes the actual roles that peer supporters assume when providing peer support and the types of strategies they implement. Specifically, these are the actions taken by peer supporters to operationalize the key components that support the benefits and outcomes of peer support. This theme provides potential roles and strategies that peers with intellectual and developmental disabilities could use to support other people with intellectual and developmental disabilities to engage in inclusive research teams.

**Table 1.** Peer support key components, roles, and strategies.

| Author(s) | Key Components of Peer Support | | | | | Peer Support Roles and Strategies | | | | |
|---|---|---|---|---|---|---|---|---|---|---|
| | Shared Lived Experience | Self-Efficacy/Self-Advocacy | Reciprocity | Friend-ship/Comradery | Role Model-ing/Education | Support for Specific Tasks | Relationship Building | Creating a Safe Space | Positive Disability Iden-tity/Normalization of Disability | Creating a Sense of Commonality |
| (Bazzano et al. 2009) | x | x | | | | | | | | |
| (Bertilsdotter Rosqvist 2019) | x | | | | | | | x | x | x |
| (Borisov 2009) | | x | x | | x | x | | | | |
| (Brady et al. 2016) | | | | | x | x | | | | |
| (Carley 2018) | x | | | x | | | | x | | |
| (Crane et al. 2021) | x | | | | | | | x | | |
| (Davis et al. 2018) | | | | | | x | | | | |
| (Eisenman et al. 2014) | x | x | | | | | | | | |
| (Frawley and Bigby 2014) | x | x | x | | x | | | | | |
| (Hillier et al. 2007) | | | | | | | x | | | |
| (Kearney et al. 2018) | | | | x | | | x | x | | |
| (Kramer et al. 2018) | x | | x | x | | | x | x | | x |
| (Marks et al. 2019) | | x | | x | x | x | | | x | |
| (Nind et al. 2021) | x | x | x | x | | | | | | |
| (Power et al. 2016) | | x | | | | | | x | | |
| (Ryan et al. 2016) | x | | x | | | | | x | | |
| (Schwartz and Kramer 2018) | x | | | x | | | | x | x | |
| (Schwartz et al. 2020) | x | | | x | x | | | x | | |
| (Singh et al. 2011) | x | | | | | | | | | |
| (Strnadová et al. 2019) | | | | | | | | | | |
| (Strnadová et al. 2014) | x | x | x | | x | | | | x | |
| (Weidle et al. 2006) | | x | | | | | | | | |
| (Williams 2015) | x | | x | | | | | | | x |
| (Williams and Porter 2017) | x | x | x | | | | | | x | |
| (Witton et al. 2017) | | | | x | x | x | | | x | |
| (Wright et al. 2019) | | x | x | | | x | | | | |

Descriptions of the roles of peer supporters in each article varied considerably, although there were a number of consistent themes across articles. Peer supporters were described as providing friendship/comradery (n = 8) (Carley 2018; Kearney et al. 2018; Kramer et al. 2018; Marks et al. 2019; Nind et al. 2021; Schwartz et al. 2020; Witton et al. 2017) and role modeling or education (n = 7) (Borisov 2009; Brady et al. 2016; Frawley and Bigby 2014; Marks et al. 2019; Schwartz et al. 2020; Strnadová et al. 2014; Witton et al. 2017). Others described peers as providing support for specific tasks (n = 8) (Borisov 2009; Brady et al. 2016; Davis et al. 2018; Kearney et al. 2018; Kramer et al. 2018; Marks et al. 2019; Witton et al. 2017; Wright et al. 2019), relationship building (n = 6) (Bazzano et al. 2009; Bertilsdotter Rosqvist 2019; Carley 2018; Crane et al. 2021; Dudley et al. 2015; Eisenman et al. 2014; Frawley and Bigby 2014; Hillier et al. 2007; Kramer et al. 2018; Schwartz et al. 2020; Singh et al. 2011; Strnadová et al. 2014; Williams and Porter 2017), creating a safe space for the peer (n = 5) (Bertilsdotter Rosqvist 2019; Carley 2018; Crane et al. 2021; Kearney et al. 2018; Schwartz and Kramer 2018), promoting the normalization of disabilities and positive disability identity (n = 5) (Bertilsdotter Rosqvist 2019; Marks et al. 2019; Strnadová et al. 2014; Williams and Porter 2017; Witton et al. 2017), and creating a sense of commonality (n = 3) (Brady et al. 2016; Schwartz and Kramer 2018; Williams 2015).

These actions were grouped during coding to identify five common roles that describe the function of the peer supporter in the peer support relationship. These roles included (1) facilitating communication, (2) sharing experiences, (3) helping peers to learn, (4) supporting peer development, and (5) creating a welcoming environment. Much of what researchers described could then be considered strategies to support each role. Strategies are the specific actions, activities, or tasks that a peer supporter can perform in their role to meet a goal. The peer supporter roles and strategies identified through the coding process are outlined in Table 2.

**Table 2.** Peer support roles and strategies.

| Role | Strategies |
|------|-----------|
| Communication | • Setting up a regular meeting time and place<br>• Communicating in ways that meet the peer's needs and preferences/choices<br>• Role modeling how to be a good communicator<br>• Advocating for the peer |
| Sharing Experiences | • Sharing stories about research and other experiences related to the intervention or targeted outcome<br>• Noticing and talking about experiences the peer supporter and peer have in common |
| Helping Peers to Learn | • Sharing resources<br>• Guiding the peer as they work to solve a problem or complete a new task |
| Peer Development | • Helping the peer set goals to grow as a researcher<br>• Helping the peer set personal and professional life goals<br>• Helping the peer keep track of their goals |
| Creating a Welcoming Environment | • Giving assurances to the peer<br>• Validating their feelings and experiences<br>• Noticing and celebrating success |

(Pfeiffer et al. 2021, https://sites.temple.edu/reachlabtemple/peer-support-manual/ (accessed on 8 August 2023)).

### 3.4. Supports and Accommodations for the Peer Supporter

The final theme extracted and analyzed within the coding process was *supports and accommodations for the peer supporter*. This theme describes the types of assistance that researchers provided to peer supporters in their research studies. While most of the studies occurred outside the context of inclusive research, similar supports and accommodations may be helpful when implementing peer support on inclusive research teams. Coding

across the articles identified two main types of assistance, including (1) individual supports and (2) environmental supports and accommodations.

Individual supports were provided by a designated person to help the peer supporter in their role (n = 12) (Borisov 2009; Davis et al. 2018; Eisenman et al. 2014; Frawley and Bigby 2014; Hillier et al. 2007; Marks et al. 2019; Ryan et al. 2016; Schwartz and Kramer 2018; Weidle et al. 2006; Williams 2015; Wright et al. 2019). This was often a more experienced person on the team who was identified as a peer mentor, co-facilitator or teacher, or group leader. This person would work alongside the peer or provide support and supervision to allow them to fulfill their role. Consistent supervision was repeatedly identified (n = 12) (Borisov 2009; Davis et al. 2018; Eisenman et al. 2014; Frawley and Bigby 2014; Hillier et al. 2007; Kramer et al. 2018; Marks et al. 2019; Ryan et al. 2016; Schwartz and Kramer 2018; Weidle et al. 2006; Williams 2015; Wright et al. 2019) as important to determine interest and comfort with their role, address problems, clarify expectations, and support self-efficacy. The peer supporter and peer often worked together to problem solve (n = 6) (Kramer et al. 2018; Marks et al. 2019; Ryan et al. 2016; Schwartz and Kramer 2018; Williams 2015; Wright et al. 2019). This included identifying any needed environmental supports.

Environmental supports and accommodations focused on aspects of the environment that are modifiable or enhanced to support engagement in research, including the environment of the research team (n = 10) (Davis et al. 2018; Kramer et al. 2018; Marks et al. 2019; Ryan et al. 2016; Schwartz et al. 2020; Schwartz and Kramer 2018; Weidle et al. 2006; Williams 2015; Witton et al. 2017; Wright et al. 2019). These supports and accommodations were provided by varying members of the research team members, including the investigators, research staff, or interventionists on the research team.

Researchers further coded individual and environmental supports and accommodations as process-oriented supports and tangible supports and accommodations. Process-oriented supports include non-physical supports, such as providing positive reinforcement or increasing the time needed to complete a task. Another example that aligns with the core components of peer support is to provide emotional supports and modeling. Tangible supports and accommodations include providing accessible materials, using visual supports, and having accessible technology. Table 3 provides a list of the specific types of processes and tangible supports extracted across articles during the coding process. This information provides lead researchers on inclusive teams with examples of possible supports and accommodations that were used in prior research for use within their own teams. Research leads can implement supports and accommodations preemptively or when needed to promote the success of peer supporters on their teams.

**Table 3.** Environmental supports and accommodations for peer supporters.

| Non-Tangible, Process Oriented Support |
| --- |
| Intentionally build opportunities for participation during all stages of the research process |
| Practice skills before applying them |
| Provide ongoing opportunities for practice/rehearsal, including refresher trainings |
| Modify concepts to increase the understandability |
| Use specific examples (e.g., provide concrete examples of abstract concepts; role plays of interpersonal skills; examples of tasks in action |
| Provide immediate feedback to the supporter about how they are doing |
| Slow the pace of instruction |
| Increase time for skills training when needed |
| Provide positive reinforcement |
| Fade assistance during interactions |
| Use communication cues (e.g., visual prompts for turn taking) |
| Provide specific education or training on different styles of communication |
| Provide reminders regarding how to positively approach peers and other coworkers |
| Provide emotional support |

**Table 3.** *Cont.*

| Tangible Supports and Accommodations |
|---|
| Intentionally build opportunities for participation during all stages of the research process<br>Use technology to support communication (e.g., online discussion boards; synchronous typed/text messaging)<br>Provide resources or handouts when available (e.g., peer mentoring handbook)<br>Use tangible tools such as flowcharts, checklists, tip sheets for specific tasks, sample scripts, and worksheets |
| Provide accessible written materials including<br>• Font and text size;<br>• Color coding;<br>• Individualized text;<br>• Abbreviated versions of text;<br>• Electronic versions of materials. |
| Use visual supports:<br>• Picture schedule;<br>• Create word clouds of key concepts;<br>• Visual timelines;<br>• Lists of rules;<br>• Note cards. |
| Provide technical support for the use of technology:<br>• Communications (i.e., email, videoconferencing);<br>• Using devices or software. |

(Pfeiffer et al. 2021, https://sites.temple.edu/reachlabtemple/peer-support-manual/ (accessed on 8 August 2023)).

Additionally, an understanding of the types of assistance used in prior research provides a foundation of individual and environmental supports for peer supporters that are engaging others with intellectual and developmental disabilities in research teams. This is important factor when building inclusive environments within research teams and optimizing the impact of peer support in that process.

## 4. Discussion

### 4.1. Translating Findings to a Peer Support Toolkit

This rapid scoping review identified the key components of peer support, roles, and strategies used by peer supporters and considerations for supports and accommodations that may be helpful when working with peer supporters with intellectual and developmental disabilities. The research team used these findings to identify ways that concepts like communication, peer assistance, sharing personal experiences, and creating a non-judgmental environment (e.g., Brady et al. 2016; Crane et al. 2021; Frawley and Bigby 2014; Schwartz et al. 2020) can be embedded in the research process. This movement toward more inclusive research teams can propel future research that is directly informed by people with intellectual and developmental disabilities. The application of peer support as a method for research engagement advances present engagement strategies that focus more on research participation by clearly identifying a central role for a person with lived experience as a member of the research team. We operationalized what we learned into the developmental of a Peer Support Toolkit (Pfeiffer et al. 2021, https://sites.temple.edu/reachlabtemple/peer-support-manual/ (accessed on 8 August 2023)).

The Peer Support Toolkit was developed in collaboration with a range of people with intellectual and developmental disabilities who worked on research teams at three institutions Temple University, University of Florida, and Boston University. Our team members had a range of experience conducting research. Team members at Temple University had 1–5 years of experience, while the team members at Temple University (who refer to themselves as the "Cool Cats") had less than one year of experience, and almost all of

their experience was asynchronous work conducted during the first year of the pandemic. One of the authors, an experienced peer supporter and autistic researcher from Temple University and the Cool Cats met weekly for several months to review the results from the rapid scoring review and identify and develop information that could be included in the toolkit. Team members at Boston University developed materials asynchronously, based on their identified interests and more limited availability during the project.

The first component of the toolkit, "For Peer Supporters", was developed with the intention to teach team members with intellectual and developmental disabilities how to serve as peer supporters on a research team. This component included two units: (1) what is peer support and (2) roles and strategies. In the sections for each unit, our collaborative team was responsible for translating key concepts into accessible language and identify images to support the meaning of the words. The team also worked on the development of interactive content to support learning, such as generating real-life examples of concepts, developing role plays, and creating videos. One of the authors created printable worksheets designed to support the use of the strategies based on worksheets and materials used by her inclusive teams, which were then trialed and refined by the Cool Cats. Table 4 includes a summary description of the materials developed by the team.

One of the most significant tasks completed by the collaborative team for this section of the toolkit was the creation of role play videos. Three videos demonstrate a peer supporter enacting three of the roles identified in the scoping review: (1) how to ask for accommodations during a research team meeting (role: help people communicate in a way that works for them); (2) coaching a new team member about how they handled a challenging situation while working with a research participant (role: help the peer mentor learn new research tasks); and (3) giving assurance to a new members of the team (role: create a welcoming environment). Videos may be more beneficial than words on paper, or adding an activity, for people with different learning needs and styles. Videos that show people with the lived experience of having intellectual and developmental disabilities engaging in research also facilitate the realization that "they are like me, and they are part of a research team, and that's something I want to be involved in" (as articulated by a member of our team). This could build self-efficacy, a key component of peer support (Bandura 2012; Burke et al. 2019; Dennis 2003)and change people's perspectives of what is possible for their career.

These video scripts were developed by the experienced peer supporter. He generated the content for the script by drawing upon his previous experience with working as a peer supporter, thinking of strategies he used as a person with a disability to participate in research, and reviewing the example roles and strategies identified during the scoping review. Table 5 includes an extended reflection from the peer supporter on the research team about the process of providing peer support to other members with intellectual and developmental disabilities on the team.

The creation of this Peer Support Toolkit was also an opportunity for the more experienced researcher to enact the roles and activate the key components identified in the peer review. The key component of reciprocity is apparent in the peer supporter's reflections, as he had the opportunity to support others and build capacity, while also being exposed to new things and building his skills and awareness (Thiele et al. 2019; Substance Abuse and Mental Health Services Administration (SAMHSA) (2015)). Working together on this project also created a sense of connection and community, and people's different disabilities and abilities were respected. The team's use of process-oriented and tangible supports and accommodations, as identified in the review, facilitated everyone's engagement in the development process, regardless of their years of research experience or communication style. These accommodations were especially crucial, since the team was working in a virtual environment (Kramer et al. 2023).

**Table 4.** Peer Support Toolkit units and materials developed to train peer supporters.

| Unit and Section | Example Contribution of Team Members with Disabilities |
|---|---|
| Unit: What is peer support? | • Lived experience videos about being members of a research team.<br>• Understanding the difference between formal and informal peer support, using pictures and examples |
| Unit: Roles and strategies for peer support | |
| Help people communicate in ways that work for them | • Provided examples of how to set up a regular meeting time and place.<br>• Thought of accommodations that could support members of a research team (e.g., use chat on Zoom meetings, use plain language).<br>• Gave examples of communication (body language, listening, telling others when you do not know, speaking up) that is "good" (e.g., ask people to slow down) and "needs improvement (e.g., yell at people).<br>• Trialed a worksheet that peer supporters could use to help other team members plan ahead about what they want to share at a meeting.<br>• Created a role play video showing a peer supporter helping a new member of a research team ask for accommodations. |
| Share your experiences | • Generated examples of things that are hard when working on a research team (e.g., lots of information is shared at a meeting) and the solutions they use to resolve the barriers (e.g., reviewing the agenda ahead of time).<br>• Activities a peer supporter could do to get to know new people on the team (e.g., play icebreakers, go to the snack bar together). |
| Help the peer researcher learn new research tasks | • Links to videos our team used to learn about research<br>• Created a role play video showing a peer supporter coaching a new member of the research team |
| Support personal and professional development | • Listed research skills that new team members may want to learn (e.g., learn how to use Excel, learn how to ask good questions during interviews)<br>• Trialed a worksheet that peer supporters could use to help other team members set personal and professional goals and keep track of their goals. |
| Create a welcoming environment | • Things a peer supporter could say to help new researchers feel confident when learning new things (e.g., "Keep trying", "I can help explain this to you", and "You are trying new things")<br>• How a peer supporter could recognize the successes of new team members (e.g., say "good job", send a text with a fun emoji)<br>• Created a role play video showing a peer supporter providing assurance to a new member of the research team |

The second component of the toolkit, "For Research Team Leads", was developed to provide team leads, with or without disabilities, with practical resources to integrate peer support into their inclusive research team. The project leads were responsible for developing the content based on research and years of experience. The first unit, "Research Engagement and Peer Support", integrates key concepts from peer support and inclusive research and provides links to external resources and publications about research collaborations with people with intellectual and developmental disabilities. It draws on supporting literature from inclusive research that identifies important concepts to make research engagement by people with lived experience more successful. These concepts include fair and equitable power between team members; trust between people with lived

experience and researchers; education and regular communication for both researchers and team members with lived experience; and adequate compensation for the time and expertise of team members with lived experience (Bigby et al. 2014; Franke et al. 2019; Harrison et al. 2019; Nind and Vinha 2014). The key components of peer support overlap with these concepts, which enhances the roles of peer support in successful engagement in inclusive research teams. The second unit, "Peer Researchers in the Phases of Research", maps the various roles of peer support to each phase of research, from choosing a research topic, to data collection, to dissemination. The third unit, "Recruiting and Hiring", draws upon our own experience with recruiting, hiring, and onboarding team members with disabilities to provide exemplar job descriptions, accessible interview procedures, and other hiring considerations.

**Table 5.** A peer supporter's reflections on providing support to an inclusive research team.

| |
|---|
| When making the scripts and videos for the tool kits, I used my personal experiences. I used to be in theater in high school, so I used my skills from theater. More importantly, our team talked about each script and made edits as a team. I like to be detailed, but not everyone is like that—so we wanted to make sure the story made sense for everyone on the team. We spent a lot of time practicing before the recording. We recorded different videos, and then picked one that we thought was the best. |
| One of the challenges doing this work was the distance between the two research teams. Temple University is almost 1000 miles away from the University of Florida. It would have been nice to come together in person and collaborate, but distance and the COVID-19 pandemic made this impossible. Differences in communication styles, and one team member's use of an Augmentative and Alternative Communication (AAC) device, was not something I had a lot of experience with. Coordinating times we are all available for our meetings was also challenging. |
| Our team had to create a welcoming environment to deal with these challenges. Our team did a good job of balancing the responsibility of meeting the deadlines for the project and working together as a team in a way that is positive and enjoyable. We worked hard, but it did not feel like a lot of pressure, because we planned ahead and had a schedule. |
| When I first started working on this project, it was something as another part of my responsibilities. I didn't realize the impact of what we did until later on. When I was no longer able to attend the meetings and started going into other spaces with very few people with disabilities, I realized the impact and uniqueness of our team. I am able to talk about my experience working with a research team that is inclusive of people with lived experience of disabilities with others. When I explain to people what we did, which is usually to neurotypical people, I try to teach other people that including peers on research team is really valuable for everyone. It provides employment to people with disabilities and it changes the perspectives of people who are not exposed to different disabilities—like autism, cerebral palsy, and other disabilities. We all have different perspectives about disabilities, and maybe they think people with disabilities can't do anything. Working with people with different disabilities helps you understand what they can do and changes perspectives. |
| I learned that I had the ability to teach these skills and be a mentor for new researchers. Especially for people who are in the same stage of life and the same age as me. I hope in the future I can meet more people who are around the same age as me—and potentially all over the country—and we can share similar experiences. |

An initial draft of the toolkit was reviewed by two researchers and two peer supporters with intellectual and developmental disabilities. All reviewers shared feedback to improve the overall layout and navigation of the toolkit pages, recommendations to improve the accessibility of the language, and the use of pictures and visual supports. Revisions based on these recommendations included a redesigned home page that described the toolkit's audience. Overall, the reviewers felt that the toolkit was useful, and they liked the worksheets. The process of toolkit development was inclusive at all stages, which is a unique aspect of this work and one that significantly improves the quality of the toolkit created (Pfeiffer et al. 2021, https://sites.temple.edu/reachlabtemple/peer-support-manual/ (accessed on 8 August 2023)).

### 4.2. Limitations

This rapid scoping review was designed to quickly identify peer support strategies that could be leveraged to facilitate research engagement with people with intellectual and developmental disabilities. Many of the articles identified in the search did not include the demographic characteristics of participants, although some studies did identify the age range. Without descriptions of race, ethnicity, gender, and other demographic information, it is impossible to determine if the studies examined are collectively representative of the population. It is possible that different features of peer support may be more effective for different communities and populations. In addition, there was limited research that focused on including peer support within research teams, and these studies did not always focus on the effectiveness of peer support. This review included all types of study design in order to broadly identify the characteristics and roles of peer support.

### 5. Conclusions

This rapid scoping review identified the components of peer support, the roles and strategies used by peer supporters, and supports and accommodations for the peer supporter. The toolkit will help people with intellectual and developmental disabilities to engage in research, creating more inclusive research teams and research informed directly by the needs of people with lived experience. Ensuring those with intellectual and developmental disabilities are included in research in a supportive way will improve the effectiveness and equitable nature of future research.

**Supplementary Materials:** The following supporting information can be downloaded at: https://www.mdpi.com/article/10.3390/socsci13010047/s1, Table S1: Matrix of Rapid Scoping Review Articles.

**Author Contributions:** Conceptualization, B.P., T.H. and J.K.; methodology, B.P. and J.K.; validation, L.T.; formal analysis, B.P. and T.H.; investigation, T.H.; resources, B.P. and J.K.; data curation, T.H.; writing—original draft preparation, B.P. and J.K.; writing—review and editing, T.H. and L.T.; visualization, B.P. and J.K.; supervision, B.P. and J.K.; project administration, B.P., T.H. and J.K.; funding acquisition, B.P. and J.K. All authors have read and agreed to the published version of the manuscript.

**Funding:** This project was funded through a Patient-Centered Outcomes Research Institute (PCORI) Eugene Washington PCORI Engagement Award (EAIN-00109): Preparing individuals with IDD for Engagement in Research during Public Health Emergencies (EAIN-00109).

**Data Availability Statement:** Details of the full search strategy for the rapid scoping review are provided at http://hdl.handle.net/20.500.12613/4663 (accessed on 21 December 2023) (Roth et al. 2020), which is a freely available repository for sharing and archiving a range of scholarly works, including search processes for reviews. Specific search methods and terms for each database are available in metadata available at this site for replication purposes.

**Conflicts of Interest:** The authors declare no conflict of interests.

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
