# Peer review of "Peer Support Provided by People with Intellectual and Developmental Disabilities: A Rapid Scoping Review to Develop a Toolkit for Inclusive Research"

_socsci, doi:10.3390/socsci13010047_

Round 1

Reviewer 1 Report

Comments and Suggestions for Authors

I can see potential for a good paper here but I have a number of concerns that I think need to be addressed. I outline these here and encourage the authors to consider them.

I would prefer not to see people referred to as people with IDD as this is somewhat objectifying/reductionistic and widely disliked by self-advocates.

The abstract needs to indicate what literature you were scoping – the focus is somewhat unclear here. The emphasis is on inclusive research but the reader needs to know from the abstract whether the scoping review was conducted in an inclusive way. There is virtually no background literature about inclusive research included in the paper and this is a significant gap.

I find the style of this paper overly certain – you state many ‘facts’ in the opening paragraphs without any consideration that statements like: ‘Inclusive research teams actively engage people with IDD in all stages of research development, implementation, and dissemination’ might be contentious with other definitions proposed in the literature. You are rather quick to assert what is essential and what must happen. Good social research, I would suggest, recognise that such realities are constructed and open to question.

There is an assumption that the shared lived experience underpinning support had to be living with intellectual and development disability but I wonder if this overly privileges this one dimension. We are after all people with intersectional identities and people can have more in common with others of the same age, gender, heritage etc than they do across a very heterogeneous disability label. Is being a ‘peer’ rather limited in your conceptual framing?

You state that the overall study goal was to quickly translate the existing knowledge into a toolkit; this is great but I am not convinced that              it forms a strong basis for an academic paper.

Methods: I think some greater transparency is needed – what was the rationale for choosing age 13 for your cut off point for inclusion? Why was an editorial included? How did you select which data to extract and how did you do your analysis?

The tables can be difficult to read, left-justifying the text would help, but they do contain useful information.

There is a leap in the paper – from peer support in a range of contexts to inclusive research. Readers (I’m guessing not just this reader) need a bridge to be built to help scaffold our way across this leap. I want to understand better how the peer support idea and evidence is adding something new to the extensive literature on inclusive research. The paper gives no indication that you are familiar with any of this literature and that erodes confidence in your argument.

The paper needs some critical reflection about its limitations at the end. Indeed a more critical and reflective approach throughout would greatly improve a paper that has real promise.

Author Response

Reviewer 1:

I can see potential for a good paper here, but I have a number of concerns that I think need to be addressed. I outline these here and encourage the authors to consider them.

I would prefer not to see people referred to as people with IDD as this is somewhat objectifying/reductionistic and widely disliked by self-advocates.

We defer to the editors of this special issue for terminology.

As noted in a recent review, the inclusive research literature coming out of the US has been relatively absent. Part of this absence may be differences in describing disability in the research literature.  In the US, our leading journals require us to use the term intellectual and developmental disabilities. Individuals with disabilities in the US do not have consensus on the language they prefer- some use the term self-advocate, while others refer to intellectual disability, developmental disability, disability, or even a specific condition (cerebral palsy, down syndrome). 

The use of language can be further problematized in research teams that collaborate with individuals with a range of conditions, such as our team which includes people identifying with all of these groups as well as a person who identifies as autistic.

The abstract needs to indicate what literature you were scoping – the focus is somewhat unclear here. The emphasis is on inclusive research but the reader needs to know from the abstract whether the scoping review was conducted in an inclusive way. There is virtually no background literature about inclusive research included in the paper and this is a significant gap.

The introduction was revised to situate the potential value of peer mentoring as a method for enhancing capacity and engagement of people with intellectual and developmental disabilities in research.

I find the style of this paper overly certain – you state many ‘facts’ in the opening paragraphs without any consideration that statements like: ‘Inclusive research teams actively engage people with IDD in all stages of research development, implementation, and dissemination’ might be contentious with other definitions proposed in the literature. You are rather quick to assert what is essential and what must happen. Good social research, I would suggest, recognise that such realities are constructed and open to question.

The introduction has remained concise and we have incorporated a range of citations to support our statements. We believe the revisions may address some of these concerns.

There is an assumption that the shared lived experience underpinning support had to be living with intellectual and development disability but I wonder if this overly privileges this one dimension. We are after all people with intersectional identities and people can have more in common with others of the same age, gender, heritage etc than they do across a very heterogeneous disability label. Is being a ‘peer’ rather limited in your conceptual framing?

We agree with this reviewers’ astute observation. In the introduction, we recognize that people bring multiple identities to their experience as a peer supporter, but situate our paper in the underrepresentation of people with intellectual and developmental disabilities as peer supporters.

Methods: I think some greater transparency is needed – what was the rationale for choosing age 13 for your cut off point for inclusion?

Information was added in the Methodology to provide more information on the cut off age for inclusion. The following statement was added:

“The age range was set at thirteen and above to ensure inclusion of articles focused on the transition age youth and young adult population with IDD, as there is a subset of the literature that specifically focuses on peer support within this age range (Ryan, Kramer & Cohn, 2016, Weidle, Bolme & Hoyland, 2006). This allowed the team to find literature that included information that could translate into peer support strategies for adults with IDD on inclusive research teams.”

Why was an editorial included?

This is a very good question. The editorial was included as it contained information and data that supports the use of peer facilitated support groups to promote inclusion written by a parent and person with lived experience. We felt this information was valuable and the supporting data was not found published elsewhere. We added in the statement “with included supporting data” when referring to the editorial in the paper.

How did you select which data to extract and how did you do your analysis?

Additional information was added in the Screening and Review section to describe the process of selecting categories for the data extraction and analysis methods.

The tables can be difficult to read, left-justifying the text would help, but they do contain useful information.

The tables were modified to justify information on the left. Table 1 was moved to an Appendix.

There is a leap in the paper – from peer support in a range of contexts to inclusive research. Readers (I’m guessing not just this reader) need a bridge to be built to help scaffold our way across this leap. I want to understand better how the peer support idea and evidence is adding something new to the extensive literature on inclusive research. The paper gives no indication that you are familiar with any of this literature and that erodes confidence in your argument.

We made substantial changes to the overall paper, which includes providing a clear link from the scoping review to inclusive research and the inclusive research literature.

Reviewer 2 Report

Comments and Suggestions for Authors

I can see the value in this article and their review. However, the authors have done a terrible job at setting up a rationale, presenting their findings and being clear about how they developed a toolkit. I found it incredibly confusing as to what the “toolkit” actually is. I recommended major revision (and not rejection) based on the fact that there is utility in their idea to the field – they just need to present the information in a way that makes sense.  

I had a minor concern about publishing a duplication of work – as they refer to themselves at odd points (e.g., search terms and table 3) but I suspect they’ve already published the toolkit on a website and are referencing it – but I would like to see the author’s response.

Author Response

Peer Reviewer 2:

The manuscript describes a review of peer support research to gather learnings around delivering peer support in an inclusive research setting. While a worthy topic, the way the manuscript is not presented in a way that tells the reader what is in the toolkit or adequately sells the importance of a toolkit.

Introduction: given this is not a disability/intellectual disability journal more background is needed around intellectual and developmental disability, the importance of inclusive research, some of the challenges of running an inclusive research project (e.g., supporting or upskilling people with intellectual and developmental disability) and why this is so important in our field. The current introduction is not sufficient to sell this concept to a non-expert reader. It would also be important to explain why you chose intellectual and developmental disability and not just intellectual disability for this review. In my experience the way I work with a co-researcher with intellectual disability is quite different to how I work with an autistic co-researcher.

There are extensive revisions in the Introduction to address these concerns. There is a whole paragraph and additional information in other paragraphs that describe IDD and the importance of stakeholder research for this specific group of people.

Methods: I was confused as to why the search terms were published elsewhere - have you published the review already?

Thank you for this question. We have added additional information in the manuscript to clarify that the search strategy and process was added into an on-line repository with metadata for replication purposes.

“To identify studies to include or consider for this rapid scoping review, the review team collaborated with a medical librarian to develop detailed and systematic search strategies for each database (Bramer, et al., 2018). Details of the full search strategy are provided at (hidden for peer review), which is a freely available repository for sharing and archiving a range of scholarly works including search processes for reviews. Specific search methods and terms for each database are available in metadata available at this site for replication purposes.”

Some of the detail in the screening/review section could be edited to be more concise.

This section was modified to make it more concise and edited for better organization and ease of reading.

Could you please explain the age range decision? Is this because more peer support interventions happen with young people/is your work involving young people more so than adults.

Information was added in the Methodology to provide more information on the cut off age for inclusion. The following statement was added:

“The age range was set at thirteen and above to ensure inclusion of articles focused on the transition age youth and young adult population with IDD, as there is a subset of the literature that specifically focuses on peer support within this age range (Ryan, Kramer & Cohn, 2016, Weidle, Bolme & Hoyland, 2006). This allowed the team to find literature that included information that could translate into peer support strategies for adults with IDD on inclusive research teams.”

It was unclear the role the researcher with intellectual and developmental disability and how involved in the process they were. This is an opportunity to demonstrate the value they added to the process.

Additional information was added to the methods section and when describing the development of the toolkit to demonstrate the role the researcher with IDD played on the team and their involvement in the process. These additional are found in both the Methodology and the Discussion sections.

Was there an analytical process you followed?

Additional information was added into the Methodology to describe a thematic analytic process that was also guided by an advisory board and inclusive research team members.

Results: the results section needs substantial revision. It feels like a section in its infancy, and it’s yet to be pushed to the next level to present a coherent and in-depth reflection of what you found. The results read as if the results/code presented themselves to you, rather than you had an intellectual role in creating them.

Revisions were completed in the results section to discuss the codes followed by the research supporting those codes. Substantial changes were made to reflect on these codes and the relevance to inclusive research based on the results.

I did not find table 1 useful or adding to the paper. It needs to edited with the reader in mind – what key information do I need from this paper? (i.e., I don’t need to know the title but the general field like dental program, social group)

This table was made into an Appendix so as not to distract from the content of the paper, but still be available if a reader is interested.

The age range for Brady et al is missing.

Age ranges in the table were added where missing. Much of the time the article/study did not provide specific age ranges but only identified participants as adults. This was clarified.

3.1 seems to be discussing the benefits of peer support but the intro to the section is talking about a definition of peer support based on these articles. Please revise to more clearly introduce this section. 

These sections were modified to include an introduction to the results that provides an overview of the results sections. 3.1 now includes Article Characteristics and how these are relevant to peer support in inclusive research.

3.2 I struggled to understand how 3.1 and 3.2 differed and how these led to the roles. the presentation of the results needs to be revised for clarify.

Results were modified to include the differences in these themes and their relevance to inclusive research.

Table 3 very informative and useful to understand the key take away points.

Table 4 has good info but perhaps a table is not the best approach to presenting this information I did not get a sense of what I can take away from the results section to apply to my own work with co-research with intellectual and developmental disability.

Additional information was added into the text describing these supports and accommodations that connects how research team leads could use these examples within inclusive research teams.

Discussion: much of this content should be moved to the intro. The first paragraph of the discussion should highlight the key findings of your review It’s unclear what the toolkit is? Is it the things in tables 3 and 4? 4.1: this is information that should have been presented earlier in the manuscript. 4.2: this is the first time the need to develop a toolkit quickly within the context of covid is mentioned. Again, this is key information that should have been presented earlier The discussion needs considerable work; I suggest looking up some guidance about the content that needs to go in a discussion.

We completed an overhaul of the Discussion. Information from the Discussion was moved to the Introduction as recommended and the Discussion was re-written with the intention of providing substantially more information on the toolkit and the inclusive research team.

Reviewer 3 Report

Comments and Suggestions for Authors

This is a nice and inspiring scoping review about peer support provided by people with IDD. The methods are clear, although I was puzzling a bit about the six articles that were excluded in the end, because they did not meet the criteria for peer support. Why were they not detected earlier in the study? Inspiring, because of the roles described and the key and support components are not difficult to apply, this all supports engagement. To me, this paper is not about the effectiveness of peer support, but about making peer support more accessible, for researchers and for people with IDD. And that is important.

Author Response

Reviewer 3:

This is a nice and inspiring scoping review about peer support provided by people with IDD. The methods are clear, although I was puzzling a bit about the six articles that were excluded in the end, because they did not meet the criteria for peer support. Why were they not detected earlier in the study?

Thank you for this feedback. We have added in additional information here that identifies that the we decided to exclude those articles that did not provide intentional peer support where direct engagement was present. We identified that these articles were excluded because they were focused on video modelling by a peer but did not have this direct engagement.

Inspiring, because of the roles described and the key and support components are not difficult to apply, this all supports engagement. To me, this paper is not about the effectiveness of peer support, but about making peer support more accessible, for researchers and for people with IDD. And that is important.

Reviewer 4 Report

Comments and Suggestions for Authors

Thanks for the opportunity to read this interesting and important paper.  Great , this really moves us all on in relation to thinking about working more inclusively and equally in our research teams. It is clearly and well written and makes the case strongly.  The idea of the toolkit is a good one and I look forward to seeing that in due course!

I suggest only a few tiny typos/changes.

Line 166 delete in

Line 219 Researchers

Line 299 call

Line 327 Public health emergency? This aspect was not mentioned in the search details earlier?

Kind regards

Author Response

Reviewer 4:

Thanks for the opportunity to read this interesting and important paper.  Great , this really moves us all on in relation to thinking about working more inclusively and equally in our research teams. It is clearly and well written and makes the case strongly.  The idea of the toolkit is a good one and I look forward to seeing that in due course!

Thank you for your supportive review. We have made the following corrections and changes based on your feedback as outlined below.

I suggest only a few tiny typos/changes.

Line 166 delete in – This was changed in the document.

Line 219 Researchers – This was changed in the document.

Line 299 call - This was changed in the document.

Line 327 Public health emergency? This aspect was not mentioned in the search details earlier? This was taken out to reflect the broader focus on the importance of inclusive research across all types of research and at all times.

Round 2

Reviewer 1 Report

Comments and Suggestions for Authors

The paper has been extensively edited to good ends. You have chosen to retain IDD and I still find this problematic. It was not 'intellectual and developmental disabilities' as a term I was objecting to but the shortening of it to IDD. I leave the editor to decide if IDD is acceptable. 

I was pleased to see some more qualifiers and sources cited in the text to make the tone less definite. However, the citation practices inappropriately include first names and initials at times. These are formatting errors could be addressed at the proofs stage

Overall this paper is much improved - tighter and more transparent in its arguments and processes.

Author Response

Response to Reviewer 1

Reviewer: The paper has been extensively edited to good ends. You have chosen to retain IDD and I still find this problematic. It was not 'intellectual and developmental disabilities' as a term I was objecting to but the shortening of it to IDD. I leave the editor to decide if IDD is acceptable. 

Response: Thank you for the clarification and feedback. We have changed the abbreviation IDD to intellectual and developmental disabilities throughout the paper.

Reviewer: I was pleased to see some more qualifiers and sources cited in the text to make the tone less definite. However, the citation practices inappropriately include first names and initials at times. These are formatting errors could be addressed at the proofs stage.

Response: We have made changes and corrections in the citations, which includes removing any first names and initials where needed.

Reviewer: Overall this paper is much improved - tighter and more transparent in its arguments and processes.

Response: Thank you for your feedback.